# Unintentional injuries and potential determinants of falls in young children: Results from the Piccolipiù Italian birth cohort

**Martina Culasso**[1]\*, **Daniela Porta**[1], **Sonia Brescianini**[2], **Luigi Gagliardi**[3], **Paola Michelozzi**[1], **Costanza Pizzi**[4], **Luca Ronfani**[5], **Franca Rusconi**[6], **Liza Vecchi Brumatti**[5], **Federica Asta**[1]

**1** Department of Epidemiology, Lazio Regional Health Service—ASL ROMA1, Rome, Italy, **2** Center for Behavioral Science and Mental Health, Istituto Superiore di Sanità, Rome, Italy, **3** Division of Neonatology and Pediatrics, Ospedale Versilia, Viareggio, AUSL Toscana Nord Ovest, Pisa, Italy, **4** Department of Medical Sciences, University of Turin, Turin, Italy, **5** Institute for Maternal and Child Health—IRCCS "Burlo Garofolo", Trieste, Italy, **6** AUSL Toscana Nord Ovest, Pisa, Italy

\* m.culasso@deplazio.it

## Abstract

### Objectives

Unintentional injuries such as falls, are particularly frequent in early childhood. To date, epidemiological studies in this field have been carried out using routine data sources or registries and many studies were observational studies with a cross-sectional design. The aims of the study are to describe unintentional injuries in the first two years of life in the Piccolipiù birth cohort, and to investigate the association between mother and children characteristics and the First Event of Raised surface Fall (FERF).

### Methods

This longitudinal observational study included 3038 children from an Italian birth cohort. Data on socio-demographic factors, socio-economic indicators, maternal health and lifestyle characteristics and child's sleeping behavior, obtained from questionnaires completed at birth, 12 and 24 months of age, were considered in the analyses as potential risk factors of FERF. Time of occurrence of FERF was analyzed using the Kaplan-Meier method. The multivariable analysis for time to event was carried out using a Cox proportional hazards model.

### Results

Falls from raised surfaces are the leading cause of unintentional injuries in the cohort with 610 (21.1%) and 577 (20.0%) cases among children during the first and second year of life, respectively. An increased risk of FERF was associated with several risk factors: maternal psychological distress (HR 1.41, 95%CI 1.10–1.81), maternal alcohol intake (HR 1.26, 95% CI 1.10–1.45), and child's sleeping problems (HR 1.28, 95%CI 1.09–1.51). Children with older aged mothers (HR 0.98, 95%CI 0.96–0.99) and living in northern Italy (HR 0.64, 95% CI 0.55–0.75) had a lower risk of FERF.

**Data Availability Statement:** Despite the Piccolipiù data cannot be publicly shared because of privacy concerns, we will be pleased to share de-identified data upon request. Data requests may be sent to

the following health istitution: ASL Roma1 Borgo
Santo Spirito 3, 00193 Rome Italy email:
dipepi@deplazio.it.

**Funding:** The Piccolipiù study was approved and
initially funded by the Italian National Centre for
Disease Prevention and Control (CCM grant 2010)
and by the Italian Ministry of Health (art 12 and
12bis D.lgs 502/92). This work was supported by
the Italian Ministry of Health, through the
contribution given to the Institute for Maternal and
Child Health IRCCS Burlo Garofolo, Trieste, Italy
(RC 12/12). Funders had no role in study design,
data collection and analysis, decision to publish, or
preparation of the manuscript.

**Competing interests:** The authors have declared
that no competing interests exist.

## Conclusion

The results of the study suggest that a higher risk of FERF is associated with socio-demo-
graphic factors, maternal characteristics and child sleeping behavior that could hinder par-
ent empowerment.

## Introduction

Unintentional injuries are a leading cause of death among children, an important health threat
and a public health issue [1]. The pattern and etiology of injuries and their outcome vary sub-
stantially within populations and across countries but worldwide, approximately 950,000 chil-
dren die every year due to unintentional injury [2]. Previous studies investigating the
circumstances leading to child mortality following injury have found that most injuries could
be prevented [3–5].

Unintentional injuries due to falls, near-drowning and burns are particularly frequent in
early childhood, even in high-income countries [6]. A population-based cohort study con-
ducted in Japan estimated that more than 60% of children aged 1.5 years were affected by
unintentional injuries in their first year of life [6]. Individual and family-related risk factors
associated to child injury from previous studies include: gender (being a male), having a large
number of siblings (3 or more children), having mothers with psychological, and/or behavioral
problems and having a young mother (aged $< = 22$ years) [7–13].

So far, epidemiological studies on unintentional injuries in children have been mainly car-
ried out using routine data sources and registries [14–17], which included only severe injuries
reported to secondary health care service, or have often used the cross-sectional design [8, 12,
18], which is generally considered of low-quality. Moreover, the contribution from birth and
child cohorts in this field was focused on school-children and adolescents [19–22]. European
studies including preschoolers have been mainly conducted in Northern countries [23–25],
and the evidence on Mediterranean countries is limited.

Piccolipiù is a prospective cohort of newborns enrolled in Italy. Since fetal and infant life
are periods of rapid development, characterized by high susceptibility to exposures, this pro-
spective cohort was set up to investigate the effects of environmental exposures, parental con-
ditions and social factors acting during pre-natal and early post-natal life on infant and child
health and development [26].

The aims of the present study are to describe unintentional injuries in early life, with a par-
ticular focus on those due to falls from a raised surface, and to identify potential risk factors
associated with the First Event of Raised surface Fall (FERF) during the first two years of life in
the Piccolipiù cohort.

## Methods

### Study population

Piccolipiù is the name of a birth cohort of 3,358 children enrolled in six maternity wards
located in five Italian cities (Florence, Rome, Trieste, Turin, and Viareggio) between October
2011 and March 2015 [26]. Women were contacted during pregnancy or at delivery and a
written informed consent form for participation was signed by both parents at enrollment.
Enrolled mothers were asked to complete a baseline questionnaire, with questions on demo-
graphics, environmental exposures, and mother's health. Additional information was obtained

either from medical records or directly from the mother within the 48 hours after delivery. Parents were then contacted at 6, 12, 24 months, 4 and 6 years after delivery and asked to fill in self-administrated questionnaires which included information on demographics, environmental exposures, lifestyle, and mother's and child health. The Piccolipiù study was approved by the Ethics Committee of the Local Health Unit Roma E, national coordinator of the project (Prot. CE/82 09/06/2011), and of each local center [27].

For the purpose of this study, only data obtained at birth (n = 3,038), at 12 months (n = 2,897) and 24 months (n = 2,751) of age were considered. The response rates were 99.4%, 87.2% and 83.8%, respectively.

## Study outcomes

Data on unintentional injuries were provided in both 12 and 24 month's questionnaires, where parents reported if their children had experienced at least one of the following injuries in the previous twelve months (multiple answers were allowed): falls from a raised surface (e.g. bed, furniture, table, chair), burns/scalds, poisoning, foreign body ingestion, road traffic injuries and other injuries. Moreover, details about circumstances of each fall from raised surface were collected; in particular where and when the fall happened, who was with the child, type/ site of injury (multiple choice answer) and the treatment required.

## Potential risk factors

A series of potential risk factors were considered in the analyses, namely: socio-demographic factors, socio-economic indicators, maternal health and lifestyle characteristics and child's sleeping behavior.

The enrollment area (center/north of Italy), child gender, maternal age when the child was born (<30, 30–34, > = 35), maternal education level, paternal occupation (employed, not employed), maternal smoking (no/yes) during pregnancy and alcohol consumption (no/yes) during pregnancy were obtained from the baseline questionnaire. Maternal educational level was classified in three categories: low (primary school), medium (secondary school) and high (university degree).

All others variables, including maternal occupation (employed, not employed), siblings (no/yes), day care attendance (no/ yes), maternal psychological distress (high distress, moderate distress, feel good), and variables concerning child sleeping patterns, were assessed at 12 months.

Moreover, the Equivalised Household Income Indicator (EHII), was used as indicator of the total disposable monthly household income at birth, standardized by household size and composition in our cohort [28]. Piccolipiù cohort data (maternal age, cohabitation status, country of birth, educational level, occupational status and occupational code; paternal/partner age, country of birth, educational level, occupational status and occupational code; and household size and tenure status) and external data from the Italian 2011 "European Union Statistics on Income and Living Conditions" (EU- SILC) survey [29, 30] were considered. The EHII was reported as a three-level categorical variable by using the 25th (933 Euro) and 75th (1,810 Euro) percentiles of total monthly disposable household income in the Italy-EUSILC survey as cut-off thresholds.

To measure maternal psychological distress, the Italian version of the 12-items General Health Questionnaire (GHQ-12) was administered at the same time as the 12 months questionnaire. This is a self-administered questionnaire aimed at detecting current levels of general (not psychotic) psychiatric morbidity, mainly in the anxiety/depression spectrum over the past two weeks. It has been used extensively in many community and hospital settings in different

countries, including Italy and showed high validity and reproducibility [31, 32]. Mothers were asked to rate the degree to which they had experienced several symptoms and/or mood states. Answers are reported according to a four-level Likert-type scale (from "not at all" to "much more than usual"). The GHQ-12 was reported on a two-level scale by collapsing the four Likert categories into two (coded 0–1) [31]. The total score was computed by summing up the single item scores. Thus, the theoretical range was 0–12, with higher values indicating more severe distress. We considered scores between 2 and 4 to identify moderate psychological distress, and > = 5 to identify severe distress. These cut-offs have previously been adopted in the literature to screen mental problems [31, 33–35].

Children's sleeping behavior was assessed considering data provided in the questionnaire: *(i)* time needed to fall asleep (a categorical variable was created: < = 30 minutes, > 30 minutes); *(ii)* where the child sleeps (a categorical variable was created: in parents' bed (co-sleeping), in a room with others, in a room alone; *(iii)* the use of a comfort object while sleeping such as a pacifier, a cuddly toy, sucking the thumb (no/yes); *(iv)* parental perception of child's sleeping behavior. The latter, was recoded into a dichotomous variable (no/yes), in which "yes" comprised both "somewhat of a problem" and "quite a problem" responses.

## Statistical analysis

Firstly, unintentional injuries from birth to 12 months and from 13 to 24 months were described as absolute and relative frequencies.

Time to occurrence of FERF was defined as the time from birth to the first raised surface fall event. In case of multiple falls, the first one was considered. Children without a documented fall from raised surface at the end of the study were censored at the date of the last available questionnaire. For example, for a child who didn't have any fall event and had the 24 months' questionnaire filled in (the end of study period or follow-up), the time to event considered was 24. The time of occurrence of FERF was analyzed using the Kaplan-Meier method.

The Log-rank test was used to compare time of occurrence of FERF and all predictive factors.

All variables described in the previous section ("Potential risk factors") were included in the univariable analysis based on Log-rank test (S1 Table) and those significant (p-value <0.05), at this first stage, were analyzed with the Chi-Square test to evaluate their independence. Risk factors significant in the univariate analysis based on Log-rank test and not correlated among each other were included in the multivariable analysis. Time to event analysis was performed using Cox proportional hazard models, without testing interaction between covariates. We ran Cox proportional hazard models considering mother-child pairs with complete information of outcome and potential risk factors (N = 2,886). We evaluated multicollinearity in the multivariable model using the variance inflation factor (VIF), considering the presence of collinearity when VIF was higher than 10. The proportional hazards assumption was assessed through scaled Schoenfeld residuals and no relevant departure was detected. Since the missing rate was lower than 10%, we handled missing as missing at random (the pattern of missingness is not related to other variables in the dataset). All analyses were conducted using STATA 12 software (StataCorp).

## Results

In Table 1 unintentional injuries in children occurring in the first and second year of life are described by leading cause.

During the first year of life, 746 out of 2,896 children (25.8%) who filled in the questionnaire at 12 months had at least one unintentional injury, for a total number of 793 injuries.

**Table 1. Leading cause of unintentional injuries during the first and second year of life.**

|  | 0–12 months (children injured = 746) | | 13–24 months (children injured = 994) | |
| --- | --- | --- | --- | --- |
|  | *N* | *%* | *N* | *%* |
| **Fall from a raised surface** | 610 | 76.9 | 577 | 50.3 |
| **Burn/scald** | 39 | 4.9 | 81 | 7.1 |
| **Poisoning** | 3 | 0.4 | 13 | 1.1 |
| **Foreign body ingestion** | 24 | 3.0 | 25 | 2.2 |
| **Road accident** | 10 | 1.3 | 15 | 1.3 |
| **Other injury** | 107 | 13.5 | 435 | 38.0 |
| **Total** | 793 | 100 | 1146 | 100 |

During the second year of life, 994 out of 2,751 children (36.1%) who filled in the questionnaire at 24 months had at least one unintentional injury, for a total number of 1,146 injuries. Falls from raised surface were the most frequent leading cause of unintentional injury, both in the first and in the second year of life (respectively 76.9% and 50.3%).

Table 2 summarizes details of falls from a raised surface in the cohort.

During the first year of life, most falls occurred after eight months (55.4%), under parental supervision (92%) and mainly from the bed (57%). During the second year of life, falls from a raised surface occurred more frequently after twenty months (40.6%) again under parents' supervision (84.7%) and mainly from the bed (34.8%). Interestingly, the percentage of children fallen in the nursery was very low (0.8% and 3.8% in the first and in the second year respectively). Moreover, the percentage of children taken to an Emergency Department or hospitalized as a result of the fall was similar in the first two years (around 22%).

Potential risk factors in the 2,886 mother-child pairs considered in the present study are reported in Table 3.

Overall, during the first two years of life, 996 out of 2886 children (35.9%) reported a FERF.

Fig 1 displays the Kaplan-Meier survival function for the probability of FERF overall. At 12 months of age 79% of children had never experienced a fall (95% CI 77.5%-80.4%), while at 24 months of age the percentage declined to 64% (95% CI 62%-65.7%).

The results of the adjusted Cox model, including the predictive variables significant in the previous univariable analysis based on Log-rank test (S1 Table), are shown in Table 4. No collinearity was observed between predictors included in our model (VIF = 1.46).

A lower risk of FERF was observed in children living in Northern Italy (HR 0.66, 95% CI 0.57–0.77), compared to central Italy. With regards to maternal age, a lower risk was found among children with mothers aged 30–34 years (HR 0.80, 95% CI 0.66–0.96) and >35 years (HR 0.71, 95% CI 0.60–0.85) compared to those with younger mothers (<30 years old). Moreover, an increased risk of FERF was observed in children with mothers with a moderate (HR 1.41, 95% CI 1.18–1.69) and high (HR 1.50, 95% CI 1.12–2.01) psychological distress. Maternal alcohol intake during pregnancy was also found to be associated with an increasing risk of FERF among children (HR 1.23, 95% CI 1.07–1.41). Furthermore, an increased risk of FERF was found among children having sleeping problems (HR 1.33, 95% CI 1.13–1.56).

## Discussion

Our study identified maternal age, maternal distress, maternal alcohol consumption, child sleeping problems and enrollment area as predictive factors of FERF.

Children whose mothers were relatively older (> = 30 years old), had a lower risk of FERF. This finding is coherent with the evidence in the literature, in particular a study conducted in

**Table 2. Description of every raised surface fall reported during the first and the second year of life.**

| | 0–12 months (raised surface falls reported = 766) | | | 13–24 months (raised surface falls reported = 706) | |
|---|---|---|---|---|---|
| | *N* | *%* | | *N* | *%* |
| **Age at fall** | | | **Age at fall** | | |
| 1–4 months | 56 | 7.4 | 13–16 months | 150 | 21.6 |
| 5–8 months | 282 | 37.2 | 17–20 months | 262 | 37.8 |
| 9–12 months | 420 | 55.4 | 21–24 months | 282 | 40.6 |
| **Place where fall happened** | | | **Place where fall happened** | | |
| Bedroom | 479 | 62.8 | Bedroom | 268 | 38.1 |
| Living room | 147 | 19.3 | Living room | 218 | 31.0 |
| Nursery | 6 | 0.8 | Nursery | 27 | 3.8 |
| Playground/garden | 31 | 4.1 | Playground/garden | 75 | 10.7 |
| Other | 100 | 13.1 | Other | 115 | 16.4 |
| **From where the child fell** | | | **From where the child fell** | | |
| Table/chair/high chair | 89 | 11.6 | Table/chair/high chair | 194 | 27.5 |
| Bed | 437 | 57.0 | Bed | 246 | 34.8 |
| Changing table | 46 | 6.0 | Changing table | 16 | 2.3 |
| Arms | 15 | 2.0 | Arms | 20 | 2.8 |
| Other (e.g. stroller, stairs, etc) | 179 | 23.4 | Other (e.g. stroller, stairs, etc) | 230 | 32.6 |
| **Who was with the child** | | | **Who was with the child** | | |
| Parents | 690 | 92.0 | Parents | 594 | 84.7 |
| Granparents/relatives | 42 | 5.6 | Granparents | 58 | 8.3 |
| Siblings | 8 | 1.1 | Siblings | 13 | 1.9 |
| Baby sitter | 10 | 1.3 | Baby sitter | 17 | 2.4 |
| Other | 0 | 0.0 | Other | 19 | 2.7 |
| **Injury type/site of injury*** | | | **Injury type/site of injury*** | | |
| Bruise | 419 | 59.8 | Bruise | 319 | 40.9 |
| Bleeding | 63 | 9.0 | Bleeding | 73 | 9.4 |
| Wound | 18 | 2.6 | Wound | 61 | 7.8 |
| Fracture | 8 | 1.1 | Fracture | 14 | 1.8 |
| Blow | 48 | 6.8 | Blow | 54 | 6.9 |
| Head injury | 14 | 2.0 | Head injury | 15 | 1.9 |
| Other | 131 | 18.7 | Nothing | 219 | 28.1 |
| | | | Other | 25 | 3.2 |
| **Treatment** | | | **Treatment** | | |
| Medical examination | 45 | 5.9 | Medical examination | 18 | 2.6 |
| Emergency Hospital Department | 152 | 20.1 | Emergency Hospital Department | 153 | 22.0 |
| Hospitalization | 16 | 2.1 | Hospitalization | 1 | 0.1 |
| Nothing | 540 | 71.3 | Nothing | 517 | 74.2 |
| Other | 4 | 0.5 | Other | 8 | 1.1 |

Total number may vary across variables due to missing values

* *question with multiple choice*

the UK reported that younger maternal age is an important risk factor for unintentional injuries among preschoolers [23]. As Mitton and colleagues hypothesized, younger mothers may be less aware of the risks that children encounter and could be more prone to injuries growing up [36].

**Table 3. Potential risk factors of the 2,886 mother-child pairs.**

| | N | % |
|---|---|---|
| **Enrollment area** | | |
| Central Italy | 1929 | 66.8 |
| Northern Italy | 957 | 33.2 |
| **Child gender** | | |
| Male | 1471 | 51.0 |
| Female | 1415 | 49.0 |
| **Maternal age at delivery** | | |
| <30 | 556 | 19.3 |
| 30–34 | 992 | 34.4 |
| > = 35 | 1,338 | 46.3 |
| **Maternal education** | | |
| Primary school | 301 | 10.4 |
| Secondary school | 1,242 | 43.1 |
| University degree or above | 1,343 | 46.5 |
| **Maternal employment** | | |
| No | 739 | 25.6 |
| Yes | 2,091 | 72.5 |
| Not respondant | 56 | 1.9 |
| **Paternal employment** | | |
| No | 84 | 2.9 |
| Yes | 2,784 | 96.5 |
| Non-responders | 18 | 0.6 |
| **EHII**[*] | | |
| Low income | 126 | 4.4 |
| Medium income | 1,607 | 55.7 |
| High income | 965 | 33.4 |
| Non-responders | 188 | 6.5 |
| **Number of siblings** | | |
| No | 1,707 | 59.2 |
| Yes | 1,175 | 40.7 |
| Non-responders | 4 | 0.1 |
| **Day care attendance** | | |
| No | 2,000 | 69.3 |
| Yes | 708 | 24.5 |
| Non-responders | 178 | 6.2 |
| **Maternal distress** | | |
| Feel good | 2,146 | 74 |
| Moderate distress | 335 | 13,0 |
| High distress | 106 | 4.1 |
| Non-responders | 299 | 10.4 |
| **Maternal smoking during pregnancy** | | |
| No | 2,261 | 78.3 |
| Yes | 623 | 21.6 |
| Non-responders | 2 | 0.1 |
| **Maternal alcohol intake during pregnancy** | | |
| No | 1,161 | 40.2 |
| Yes | 1,667 | 57.8 |

(*Continued*)

**Table 3.** (Continued)

|  | *N* | *%* |
|---|---|---|
| Non-responders | 58 | 2.0 |
| **Time needed to fall asleep** | | |
| < = 30 min | 2,576 | 89.3 |
| >30 min | 221 | 7.7 |
| Non-responders | 89 | 3.0 |
| **Sleeping problems (reported by parents)** | | |
| No | 2,341 | 81.1 |
| Yes | 480 | 16.6 |
| Non-responders | 65 | 2.3 |
| **Where the child sleeps** | | |
| Own bed | 636 | 22 |
| Bed in a room with orthers | 1,554 | 53.9 |
| Parents bed (Cosleeping) | 584 | 20.2 |
| Non-responders | 112 | 3.9 |
| **The child uses a comfort object for sleeping** | | |
| No | 1,005 | 34.8 |
| Yes | 1,820 | 63.1 |
| Non-responders | 61 | 2.1 |

Non-responders indicates an unanswered question

* low income = <933 euro, medium income = 934–1809, high income = > = 1810 euro

Regarding maternal psychological distress, our study found that children with mothers with the highest GHQ-12 score are at higher risk of FERF. Factors affecting the association between maternal psychological health or depression and unintentional injury in children of pediatric age (0–3 years) have already been investigated in two longitudinal studies conducted in the UK and US [12, 25]. One possible explanation linking maternal depression and children's risk of injury is that chronically depressed mothers do not appropriately safeguard the physical environments children engage in [37, 38]. Furthermore, Phelan and colleagues reported that child supervision behavior on children aged 0–36 months differed between mothers suffering from depression and those who do not [39]. In contrast to these findings, a study conducted in UK in a deprived setting, found that maternal depressive symptoms, stress and a lack of social support do not influence the adoption of safety practices (such as smoke alarms, fireguards, safe storage of sharp objects/medicines, stair gates and window locks) [40].

Maternal alcohol consumption was associated with an increased risk of their child having a FERF occurring in the first two years of life in our study. This finding is coherent with previous studies where parental alcohol consumption has been found to increase the risk of unintentional injuries [18, 41].

Children with sleeping problems were found at higher risk of FERF compared to children with no sleeping problems. Sleeping difficulties in children might be correlated to sleep deprivation in parents. Considering that parental supervision plays a critical role in maintaining child safety, the consequence of parental sleep deprivation might be indicative of a less attentive or effective supervision of their children [42, 43].

The geographical differences in the risk of FERF observed in our study is somewhat difficult to explain with the data at hand and requires further analysis to evaluate if safety behaviors and risk perceptions differ by regional, social and cultural setting in Italy.

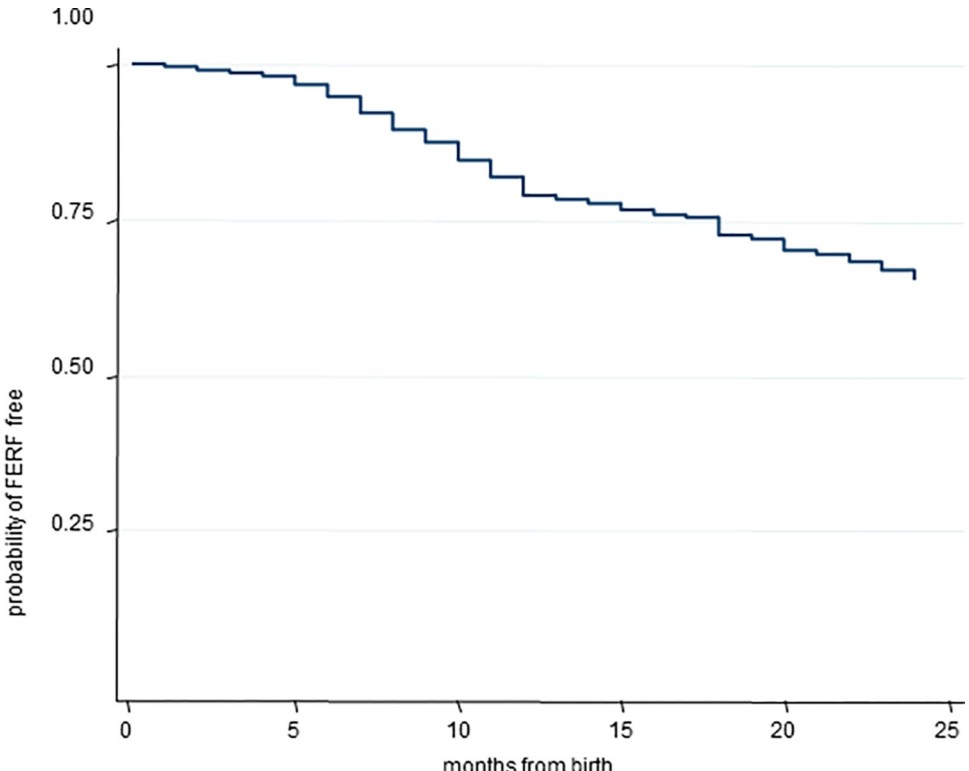

**Fig 1. Kaplan Meier survival function for the probability of first fall from a raised surface overall during the first two years of life.** FERF = First Event of Raised surface Fall.

**Table 4. Estimated hazard ratios (HRs) for FERF from the multivariable Cox model.**

| | | FERF 0–24 months | |
|---|---|---|---|
| | | *HR* | *CI 95%* |
| **Enrollment site** | | | |
| | Central Italy | 1,00 | |
| | Northern Italy | 0.66 | 0.57–0.77 |
| **Maternal age** | | | |
| | <30 | 1,00 | |
| | 30–34 | 0.80 | 0.66–0.96 |
| | > = 35 | 0.71 | 0.60–0.85 |
| **Maternal distress** | | | |
| | Feel good | 1,00 | |
| | Moderate distress | 1.41 | 1.18–1.69 |
| | High distress | 1.50 | 1.12–2.01 |
| **Maternal alcohol intake** | | | |
| | No | 1,00 | |
| | Yes | 1.23 | 1.07–1.41 |
| **Child sleeping problems** | | | |
| | No | 1,00 | |
| | Yes | 1.33 | 1.13–1.56 |

No association was found between FERF and factors such as maternal education, parental employment (a proxy of availability for supervision of the child at home), having siblings, EHII, maternal smoking during pregnancy (which could indicate a lower attention to the needs and health of the child), nursery attendance, thus we were not able to give strength to the role of "moderation" of welfare factors in injuries, as previously shown by other authors [17].

Although it is well known that boys experience injuries and falls more often than girls, this study did not show any difference by gender [9–11, 44]. Although the mechanisms through which this disparity may arise are not entirely clear, it has been suggested that boys are generally more active than girls thus are more likely to incur in injuries at an earlier age [44].

Furthermore, it is important to recall that our analysis included FERF occurring during the first year of life in which child-related factors may be less important compared to mother-related factors, potentially having a role in explaining the null association found between gender and FERF.

Several limitations of this study should be mentioned. First of all, since all the information about unintentional injuries were collected from self-reported questionnaires, parents can intentionally or unintentionally underreport injuries that children have experienced; a questionnaire administrated by a properly trained interviewer, might have reduced the risk of bias and ensure a higher quality of data, reducing definitely the amount of missing data. A social desirability bias may exist so that fall injuries can be underreported; moreover, a recall bias due to a relatively long time from injury to questionnaires may occur [45, 46]. Second, collected data included maternal and child factors, but not environmental factors (such as baby gates, window guards, restraining strap use, etc.) or other factors related to the kinetic energy on impact (such as fall height and cushioning capacity of the surface of impact).

Finally, since this study considered only the first event of fall from raised surface, future studies could be focused to assess multiple and repeated falls and evaluated if risk factors are confirmed or differ.

## Conclusion

Despite the limitations mentioned above, the present study represents a crucial investigation regarding the predictive factors of FERF in children, simultaneously explored in early life, and adds evidence in this field of research, where the role of birth cohorts is limited.

Results of this study suggest that unintentional injuries in early life can be addressed by interventions and policies that target supervision of the child, especially during pregnancy and toddlers' early life, when parental role is critical to prevent childhood injury.

Further investigations will be essential to strengthen these findings, by means of which policy makers and health professionals could design prevention strategies to empower parents and significantly reduce unintentional injuries in early life.

## Supporting information

**S1 Table. Results of the Log-rank test.**
(PDF)

## Acknowledgments

Our thanks go to all the families who took part in this study and to the whole Piccolipiù team which includes the following research scientists and computer/laboratory technicians: Paola Lorusso, Maria Gabellieri, Valentina Ziroli, Valentina Colelli, Chelo Salatino, Silvia Narduzzi,

Sara Fioravanti, Giulia Poggesi, Veronica Montelatici, Antonella Ranieli, Maura Bin, Veronica Tognin, Assunta Rasulo, Laura Fiorini.

We also thank Annarita Vestri and Alessandra Spagnoli for the support provided on the statistical analyses, and Francesca de'Donato and Ursula Kirchmayer for the revision of English language. Lastly, special thanks to Manuela De Sario who helped us in revising the manuscript.

## Author Contributions

**Conceptualization:** Martina Culasso.

**Data curation:** Daniela Porta, Sonia Brescianini, Luigi Gagliardi, Paola Michelozzi, Luca Ronfani, Franca Rusconi.

**Formal analysis:** Martina Culasso, Federica Asta.

**Investigation:** Martina Culasso.

**Methodology:** Martina Culasso, Federica Asta.

**Project administration:** Daniela Porta.

**Software:** Martina Culasso.

**Supervision:** Martina Culasso, Daniela Porta, Sonia Brescianini, Luigi Gagliardi, Paola Michelozzi, Costanza Pizzi, Luca Ronfani, Franca Rusconi, Liza Vecchi Brumatti, Federica Asta.

**Writing – original draft:** Martina Culasso.

**Writing – review & editing:** Martina Culasso, Daniela Porta, Sonia Brescianini, Luigi Gagliardi, Paola Michelozzi, Costanza Pizzi, Luca Ronfani, Franca Rusconi, Liza Vecchi Brumatti, Federica Asta.

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
