## [Decision Letter · Decision Letter 0]

2 May 2022

PONE-D-21-37522Unintentional injuries and potential determinants of falls in young children: results from the Piccolipiù Italian birth cohortPLOS ONE

Dear Dr. culasso,

Thank you for submitting your manuscript to PLOS ONE. After careful consideration, we feel that it has merit but does not fully meet PLOS ONE’s publication criteria as it currently stands. Therefore, we invite you to submit a revised version of the manuscript that addresses the points raised during the review process.

We look forward to receiving your revised manuscript.

Kind regards,

Angela Lupattelli, PhD

Academic Editor

PLOS ONE

Journal Requirements:

Additional Editor Comments:

Dear authors, please describe more thoroughly how the multivariable model was built after selection of variables in the univariate analysis. This procedure is not fully described and it is difficult to reproduce by others as it stands now. It is unclear what the criteria for retaining and removing variables from the multivariable model were; similarly, no test for predefined interaction terms (if any) are mentioned. Please also reconsider the use of a p-value 0f 0.05 for variable selection in the univariate analysis; by using this threshold of significance, you may lose important information at the stage of variable selection. Please refer to the Hosmer, Applied Logistic Regression, as example foe how predictor model building should be performed. It is also unclear how multicollineraity was evaluated; it seems that was examined between covariates, before model building - this step should be clearly described.

Missing values on covariate: please indicate the overall extent of missing values on covariates in the study, and indicate how they were handled in the analysis. It seems that a listwise deletion approach was used. If this is the case, please acknowledge the limitation of having such an approach on your results as compared to more robust methods such as multiple imputation (depending on the assumption as to why data are missing).

Reviewers' comments:

Reviewer's Responses to Questions

**Comments to the Author**

1. Is the manuscript technically sound, and do the data support the conclusions?

Reviewer #1: Partly

Reviewer #2: Yes

2. Has the statistical analysis been performed appropriately and rigorously? 

Reviewer #1: Yes

Reviewer #2: Yes

3. Have the authors made all data underlying the findings in their manuscript fully available?

Reviewer #1: Yes

Reviewer #2: Yes

4. Is the manuscript presented in an intelligible fashion and written in standard English?

Reviewer #1: Yes

Reviewer #2: Yes

5. Review Comments to the Author

Reviewer #1: It is valuable to build high-quality cohorts to support examining important research questions for injury prevention. Nevertheless, however, this research did not justify several critial issue this their injury cohort.

1. It seems that the authors did not adequately justify the rationale of implementing the Piccolipiù Italian birth cohort. Specifically, it is unclear the innovations of this cohort compared to the published cohorts related to injury prevention. Consequently, the research questions to be addressed by the Piccolipiù Italian birth cohort are unclear.

2. The raised surface fall (FERF) was not clearly defined in the manuscript.

3. The selection of potential risk factors and their operational definitions was not clearly described. This is particularly important for addressing high-quality research questions.

4. In addition, it is hard to understand the "raised surface falls reported" among children within 6 months. Because we know such young children cannot independently walk. Please explain the reason for it.

Reviewer #2: This paper describes unintentional injuries and risk factors of falls in young children from a Piccolipiù Italian birth cohort. This paper addresses an important topic in a niche population. However, it would benefit from restructuring, strengthening the introduction, distinguishing the results and the discussion, and carefully reading through for any grammatical and clarity concerns.

Abstract

• Include any hypothesized findings

Introduction

• The introduction starts off strong but weakens near the end. The structure somewhat falls apart and some of the implications of the study are lost.

• Lines 58-60 on page 3 are a little confusing and could be reworded.

• Why are we specifically describing injuries in this cohort and in Italy? I think the introduction would benefit for more context on why this study is relevant and how it can be helpful to this population.

Methods

• I’m a little confused about the meaning of “Piccolipiù.” It’s described as a “birth cohort” but what does that mean in this context? At first I though it was a city in Italy but that does not seem accurate either. Further clarity on this is essential to really understand the population of this paper.

• Do you have any examples that can be provided regarding the details about the raised surface falls?

• Do you have any psychometrics to report on the EHII?

• On page 5, the line about children’s sleeping behavior (lines 121-123) is confusing. And why did it get dichotomized?

• Lines 128-134 feel out of place in the methods section. I recommend moving them to the Statistical Analysis section.

Statistical Analysis

• Lines 137-142 should be placed in the Methods section.

• I don’t understand what is meant by the following: “Children without documented falls at the end of the study were censored at the date of the last available questionnaire.” What does “censored” mean in this context?

Results

• Page 7, lines 152-153 need to be revised as they are currently confusing: “Unintentional injuries occurred to children of the Piccolipiù cohort in the first and in the second year of life are presented in Table 1.”

• For Table 2, why were some categories multiple choice and not others?

• This section overall needs improved set up and structure. It is difficult to follow the separate ideas because of the current presentation.

Discussion

• Why might younger mothers be at higher risk? There is no interpretation of this finding.

• Any thoughts on the contrast in findings listed on page 12 lines 118-221? Also, is that supposed to be a separate paragraph? It seems randomly placed.

• I am not sure what is meant for lines 231-232 (page 13), can you elaborate further?

• The limitations section can benefit from adding what can be done differently next time and/or why the limitation existed to begin with.

• Consider adding some of the reported findings currently in the discussion to the results section. Right now the discussion feels too much like the results.

Conclusion

• Could benefit from a comment on any intervention being done to address these concerns.

6. PLOS authors have the option to publish the peer review history of their article (what does this mean?). If published, this will include your full peer review and any attached files.

Reviewer #1: No

Reviewer #2: No

---

## [Author Response · Author response to Decision Letter 0]

8 Jul 2022

Editor Comments:

Dear authors, please describe more thoroughly how the multivariable model was built after selection of variables in the univariate analysis. This procedure is not fully described and it is difficult to reproduce by others as it stands now. It is unclear what the criteria for retaining and removing variables from the multivariable model were; similarly, no test for predefined interaction terms (if any) are mentioned. Please also reconsider the use of a p-value 0f 0.05 for variable selection in the univariate analysis; by using this threshold of significance, you may lose important information at the stage of variable selection. Please refer to the Hosmer, Applied Logistic Regression, as example foe how predictor model building should be performed. It is also unclear how multicollineraity was evaluated; it seems that was examined between covariates, before model building - this step should be clearly described.

RE: Dear editor, we want to thank the editor and the reviewers for the useful suggestions that we believe significantly increased the quality of the Manuscript. Regarding the issue of selection of the predictor variables considered for the multivariable model we preferred to test each potential predictor at a time, before to include it into the multivariable model. In the first step we aimed to select the predictors with a stronger association with our outcome using a log-rank test with a p-value <0.05 as inclusion criteria, since we are in a Cox model setting. About the choice of the threshold of significance, we preferred to follow a conservative approach above all, because of the big number of predictors that we had (as reported in ‘Potential risk factors’ paragraph) in order to build a Cox regression model that included the minimum set of predictors thus avoiding over adjustment and multicollinearity. Moreover, since there is not a gold standard for the cut-off to identify predictor as “statistically significant” associated to the outcome, we think this cut-off (p<0.05) was able to discriminate among many competing hypotheses, selecting those more compatible with the data. (Greenland et al. 2016, doi: 10.1007/s10654-016-0149-3; Miller et al. 2019, https://doi.org/10.1371/journal.pone.0208631).

In the second step we tested for independence among predictors, resulted significant in the first step, using a Chi-Square test since all predictors were qualitative. In the third step we run the explicative multivariable Cox proportional hazards model including all predictors significantly associated to the outcome and independent among each other. In the final model we did not use a stepwise method for retaining or removing variables and we did not test interaction terms between covariates. 

To check the presence of multicollinearity in the final model we used the variance inflation factor (VIF), considering the presence of collinearity when VIF was higher than 10.

We modified accordingly the Method section in the lines 144-157.

Missing values on covariate: please indicate the overall extent of missing values on covariates in the study, and indicate how they were handled in the analysis. It seems that a listwise deletion approach was used. If this is the case, please acknowledge the limitation of having such an approach on your results as compared to more robust methods such as multiple imputation (depending on the assumption as to why data are missing).

RE: As for missing values of covariates we reported the “non- responders” in the Table 3. Since the percentages of the missing values in the covariates were between 0.1% and 6.5% (with the only exception of a complex index of maternal distress with a missing value of 10%), we used a complete-case approach. 

We added accordingly a sentence on this aspect at page 7 line 155-157.

Reviewers' comments:

Reviewer #1: It is valuable to build high-quality cohorts to support examining important research questions for injury prevention. Nevertheless, however, this research did not justify several critial issue this their injury cohort.

1. It seems that the authors did not adequately justify the rationale of implementing the Piccolipiù Italian birth cohort. Specifically, it is unclear the innovations of this cohort compared to the published cohorts related to injury prevention. Consequently, the 

research questions to be addressed by the Piccolipiù Italian birth cohort are unclear.

RE: We thank the reviewer for this observation. We better clarified that epidemiological injury research has mostly used routine data sources and registries. Moreover, some work has used birth or child cohorts which, however, were mainly performed in Northern European countries (lines 60-68).

2. The raised surface fall (FERF) was not clearly defined in the manuscript.

RE: FERF is an acronym for “First Event of Raised Surface Fall” during the first two years of life; we have now written the acronym in capital letters (line 71). 

3. The selection of potential risk factors and their operational definitions was not clearly described. This is particularly important for addressing high-quality research questions.

RE: Regarding the selection of the predictor variables considered for the multivariable model we preferred to test each potential predictor at a time, before to include it into the multivariable model. In the first step we aimed to select the predictors with a stronger association with our outcome using a log-rank test with a p-value <0.05 as inclusion criteria, since we are in a Cox model setting. In the second step we tested for independence among predictors, resulted significant in the first step, using a Chi-Square test since all predictors were qualitative. In the third step we run the explicative multivariable Cox proportional hazards model including all predictors significantly associated to the outcome and independent among each other.

According to the request of the reviewer, the selection of predictor variables included in this study have been described in more details (lines 144-157).

4. In addition, it is hard to understand the "raised surface falls reported" among children within 6 months. Because we know such young children cannot independently walk. Please explain the reason for it.

RE: Falls are the most common mechanism of unintentional injury but of course injury’s mechanisms change with age. In this study falls from raised surface that affected children in the first and second year of life were described. In Table 2 (line 170) it is reported the place (eg. bed, chair, table...) from where the child fell in the first 12 months of life (period in which we can assume that the child is not yet able to independently walk). Results show that falls from a raised surface occurred more frequently from the bed (57%), followed by table/chair/high chair (11.6%), changing table (6%) and etc.

Reviewer #2: This paper describes unintentional injuries and risk factors of falls in young children from a Piccolipiù Italian birth cohort. This paper addresses an important topic in a niche population. However, it would benefit from restructuring, strengthening the introduction, distinguishing the results and the discussion, and carefully reading through for any grammatical and clarity concerns.

RE: We thank the reviewer for pointing this out. We have accepted your suggestions and the final manuscript was revised by a native English speaker.

Abstract

• Include any hypothesized findings 

RE: We are not sure to have understand this comment; anyway, we added a paragraph in the Abstract to clarify why unintentional injuries are important to be investigated in birth cohorts such as Piccolipiù (lines 22-25).

Introduction

• The introduction starts off strong but weakens near the end. The structure somewhat falls apart and some of the implications of the study are lost.

RE: We thank the reviewer for this comment. We modified the text according to the suggestions (lines 60-68).

• Lines 58-60 on page 3 are a little confusing and could be reworded.

RE: We thank the reviewer for these comments. We modified the text according to the suggestions (lines 60-68).

• Why are we specifically describing injuries in this cohort and in Italy? I think the introduction would benefit for more context on why this study is relevant and how it can be helpful to this population.

RE: We thank the reviewer for this comment. The Introduction has been enriched with more information from the literature available in this field (lines 60-68).

Methods

• I’m a little confused about the meaning of “Piccolipiù.” It’s described as a “birth cohort” but what does that mean in this context? At first I though it was a city in Italy but that does not seem accurate either. Further clarity on this is essential to really understand the population of this paper.

RE: We thank you for this comment. We clarified in the text that Piccolipiù is the name of our birth cohort (line 75).

• Do you have any examples that can be provided regarding the details about the raised surface falls?

RE: In this study, falls from raised surface that affected children in the first and second year of life were described. In Table 2 were reported some details about the age of child, the place where the fall happened (bedroom, living room, nursery etc),the place from where the child fell (bed, table/chair/high chair, changing table, arms, etc), the person who was with the child (mother, father, grandparents, siblings, baby sitter, etc), the type of injury reported (bruise, bleeding, blow, etc) and the treatment required (medical examination, emergency hospital department, hospitalization, etc). 

• Do you have any psychometrics to report on the EHII?

RE: The Equivalised Household Income Indicator (EHII), was used as indicator of the total disposable monthly household income at birth standardized by household size and composition. It was derived using Piccolipiù cohort data (maternal age, cohabitation status, country of birth, educational level, occupational status and occupational code; paternal/partner age, country of birth, educational level, occupational status and occupational code; and household size and tenure status) and external data from the Italian 2011 “European Union Statistics on Income and Living Conditions” (EU- SILC) survey. This is definitely an interesting aspect but this is not the focus of this article.

• On page 5, the line about children’s sleeping behavior (lines 121-123) is confusing. And why did it get dichotomized?

RE: Children’s sleeping behavior was assessed by considering several items of the questionnaire. One of them is the parental perception of child’s sleeping behavior which was a categorical variable in the questionnaire with 3 levels. We decided to recode this variable in a dichotomous way (no/yes), collapsing in “yes” the categories 2=”somewhat a problem” and 3=”quite a problem”, since they had low frequencies.

• Lines 128-134 feel out of place in the methods section. I recommend moving them to the Statistical Analysis section.

RE: We thank the reviewer for pointing this out. We deleted this paragraph and reworded it with a detailed explanation of the entire process of variables selection in the Statistical Analysis section (lines 144-149).

Statistical Analysis

• Lines 137-142 should be placed in the Methods section.

RE: These lines are already in the Method Section (sub-section “Statistical Analysis”). We think they are important to understand the type of analysis performed and how the model was built. We prefer to maintain this part in the same place.

• I don’t understand what is meant by the following: “Children without documented falls at the end of the study were censored at the date of the last available questionnaire.” What does “censored” mean in this context?

RE: Censoring is referred to subjects who have not achieved any fall from raised surface. For example, if a child didn’t have any event of fall and had a questionnaire filled in at 24 months (the end of study period or follow-up), the time to event considered was 24.

Results

• Page 7, lines 152-153 need to be revised as they are currently confusing: “Unintentional injuries occurred to children of the Piccolipiù cohort in the first and in the second year of life are presented in Table 1.”

RE: We slightly modified this sentence to make it clearer and more understandable (lines 160-161).

• For Table 2, why were some categories multiple choice and not others?

RE: The questionnaires at 12 and 24 months of children’s age included some specific questions regarding falls from raised surface which occurred in the first and in the second year. For all questions only one answer was admitted, except for the question related to the type of injury (since many children had more than one type of injury), as reported in Table 2.

• This section overall needs improved set up and structure. It is difficult to follow the separate ideas because of the current presentation.

RE: We agree with the reviewer and we added some sentences in order to improve this section (lines 160-161, 164, 166, 181, 193-195).

Discussion

• Why might younger mothers be at higher risk? There is no interpretation of this finding.

RE: We hypothesize that younger mothers could have a lower risk perception and therefore they could pay lesser attention to prevent children’s unintentional injuries. We explained better this concept in lines 212-214.

• Any thoughts on the contrast in findings listed on page 12 lines 118-221? Also, is that supposed to be a separate paragraph? It seems randomly placed.

RE: We thank the reviewer for pointing this out. We tried to highlight that these different findings, obtained in a study conducted in UK, could be due to a different study population, since Mulvaney and Kendrick [ref 32], differently from our study which have been performed in general population, enrolled only mothers living in socio-economically disadvantage areas (line 226).

• I am not sure what is meant for lines 231-232 (page 13), can you elaborate further?

RE: We found that children living in northern cities of Italy (Turin and Trieste) are at lower risk of FERF compared to those living in central cities (Rome, Florence and Viareggio). This result is somewhat difficult to explain with the data at hand but we could hypothesize that this could be attributable to different safety behavior of parents who live in different cultural contests (lines 239-242)

• The limitations section can benefit from adding what can be done differently next time and/or why the limitation existed to begin with.

RE: We think the self-collected data used in the study could be one of the major limitations due to high risk of information bias (eg. underreporting of injuries) according to child’s health, family characteristics and environmental variables (eg. educational level, parity, household density). Moreover, a questionnaire administrated by a properly trained interviewer could definitely reduce the amount of missing data. We added a paragraph in lines 258-260.

• Consider adding some of the reported findings currently in the discussion to the results section. Right now the discussion feels too much like the results.

RE: We agree with the reviewer and have shortened the discussion section.

Conclusion

• Could benefit from a comment on any intervention being done to address these concerns.

RE: We agree with the reviewer and we better clarified the kind of interventions could be done to reduce the unintentional injuries in early life. Since the role of parents in maintaining toddler’s safety is crucial, we think these interventions should especially promote environmental safety and supervision of the child, therefore supporting parents from the birth or during birth classes (lines 274-276).

---

## [Decision Letter · Decision Letter 1]

21 Jul 2022

PONE-D-21-37522R1Unintentional injuries and potential determinants of falls in young children: results from the Piccolipiù Italian birth cohortPLOS ONE

Dear Dr. culasso,

Thank you for submitting your manuscript to PLOS ONE. After careful consideration, we feel that it has merit but does not fully meet PLOS ONE’s publication criteria as it currently stands. Therefore, we invite you to submit a revised version of the manuscript that addresses the points raised during the review process.

We look forward to receiving your revised manuscript.

Kind regards,

Angela Lupattelli, PhD

Academic Editor

PLOS ONE

Reviewers' comments:

Reviewer's Responses to Questions

**Comments to the Author**

1. If the authors have adequately addressed your comments raised in a previous round of review and you feel that this manuscript is now acceptable for publication, you may indicate that here to bypass the “Comments to the Author” section, enter your conflict of interest statement in the “Confidential to Editor” section, and submit your "Accept" recommendation.

Reviewer #1: All comments have been addressed

Reviewer #2: (No Response)

2. Is the manuscript technically sound, and do the data support the conclusions?

Reviewer #1: Partly

Reviewer #2: Partly

3. Has the statistical analysis been performed appropriately and rigorously? 

Reviewer #1: Yes

Reviewer #2: Yes

4. Have the authors made all data underlying the findings in their manuscript fully available?

Reviewer #1: Yes

Reviewer #2: Yes

5. Is the manuscript presented in an intelligible fashion and written in standard English?

Reviewer #1: No

Reviewer #2: No

6. Review Comments to the Author

Reviewer #1: Thanks for the replies to my comments. However, I don't think the replies and revisions towards comments 1 and 3 that I raised are adequately addressed. In this case, the innovations of this paper look to be weak.

Reviewer #2: This paper remains to be critical and meaningful to the unintentional injury literature, but there are still some issues with the quality of writing and structure of the paper when considering clarity, reproducibility, and transparency about the data and statistical analyses.

Introduction

• The introduction could still benefit from added context about the significance of the research questions and research population. Please reevaluate the current literature included in the introduction and consider a section that focuses on the regions of interest.

• There remains some issues with clarity. Although I appreciate the effort put into improving lines 58-60 on page 3, it continues to be difficult to follow. It is clear that this is an important point so please take the time to revise and improve clarity.

Methods

• I appreciate the authors’ considerations of revisions for this section and especially am grateful for the additional information provided within the statical analysis section. However, there are some big issues that continue to remain.

o The sleeping behavior paragraph continues to be difficult to follow. The response to the reviewers was helpful and should potentially be integrated into the text.

o Although I understand the EHII is not the focus of the paper, given that it is not a standardized measure, it is relevant and critical to include psychometric properties to help the readers understand the validity and reliability of the measure.

o I appreciate the clarification of “censored” but do not agree it is the best way to describe what happens to those participants. Please review what you wrote in the response to the reviewer and consider if it can be better integrated in the text.

Results

• I appreciate the attention provided by the reviewers toward revisions in this section but believe it would benefit from additional revisions. Regarding the differences in questionnaires and items (e.g, some were multiple choice and some were not), the authors might consider explaining this more clearly earlier on in the methods so that it can be more easily understood in the results.

• Overall, the structure of this section is much more easy to follow.

Discussion

• Although I appreciate the interpretation about young mothers, I want to know why this may be the case. Do you have any citations to demonstrate young mothers might have lower risk perceptions? Or any theory as to why? I know this is not the main purpose of the paper but when considering interpretations provided in the discussion, think about how to best explaining the findings in your results because that information is most helpful to the readers. You do a great job of this with the maternal psychological distress comments.

• Lines 243-247 are very difficult to follow, please revise for clarity.

• The authors did a good job of addressing revisions to the limitations and conclusion sections.

7. PLOS authors have the option to publish the peer review history of their article (what does this mean?). If published, this will include your full peer review and any attached files.

Reviewer #1: No

Reviewer #2: No

---

## [Author Response · Author response to Decision Letter 1]

8 Sep 2022

Comments to the Author

5. Is the manuscript presented in an intelligible fashion and written in standard English?

Reviewer #1: No

Reviewer #2: No

RE: As suggested the entire text of the article was revised by a native English speaker. 

6. Review Comments to the Author

Reviewer #1: Thanks for the replies to my comments. However, I don't think the replies and revisions towards comments 1 and 3 that I raised are adequately addressed. In this case, the innovations of this paper look to be weak.

RE: We thank the reviewer for this observation. We tried to improve the Introduction section in order to clarify that our cohort could contribute to increase the knowledge in this field of research (lines 63-69). We have therefore also reviewed the literature, adding several new references.

Regarding the third comment previously mentioned, to define the set of potential predictors to be included in the multivariable model, we considered some a priori risk factors based on literature (such as: sex, maternal age, maternal distress) and some risk factors based on biological plausibility, for example child sleeping behavior. All risk factors included in this study were assessed trough questionnaires.

Reviewer #2: This paper remains to be critical and meaningful to the unintentional injury literature, but there are still some issues with the quality of writing and structure of the paper when considering clarity, reproducibility, and transparency about the data and statistical analyses.

Introduction

• The introduction could still benefit from added context about the significance of the research questions and research population. Please reevaluate the current literature included in the introduction and consider a section that focuses on the regions of interest.

RE: We thank the reviewer for this observation. We tried to improve the Introduction section in order to clarify why our cohort could contribute to increase the knowledge in this field of research (lines 63-69). We have therefore also reviewed the literature, as suggested, adding several new references.

• There remains some issues with clarity. Although I appreciate the effort put into improving lines 58-60 on page 3, it continues to be difficult to follow. It is clear that this is an important point so please take the time to revise and improve clarity.

RE: We thank the reviewer for this observation. We better clarify this paragraph in lines 63-69.

Methods

• I appreciate the authors’ considerations of revisions for this section and especially am grateful for the additional information provided within the statical analysis section. However, there are some big issues that continue to remain.

o The sleeping behavior paragraph continues to be difficult to follow. The response to the reviewers was helpful and should potentially be integrated into the text.

We thank the reviewer for this comment and we added an explanation in the text as suggested (lines 141-143).

o Although I understand the EHII is not the focus of the paper, given that it is not a standardized measure, it is relevant and critical to include psychometric properties to help the readers understand the validity and reliability of the measure.

RE: We thank the reviewer for pointing this out. We have not understood the meaning of psychometric properties. However, the EHII is a measure of the total disposable household monthly income equivalised for household size and composition (currency: log-euro). 

This income measure is obtained with a prediction model applied to the EUSILC data (Italian EUSILC survey for Piccolipiù) including as predictors the following variables (maternal age, maternal cohabitation status, maternal country of birth, maternal educational level, maternal occupational status and maternal occupational code; paternal age, paternal country of birth, paternal educational level, paternal occupational status and occupational code, household size and tenure status). This income measure has been derived for approximately 20 European birth cohort studies within the framework of the H2020 LifeCycle project. As described in the paper by Pizzi et al, the prediction models for each study have been already validated (temporal validation) and the derived measures compared with other SEP-related questionnaire-based information, including self-reported income, showed a strong correlation.

o I appreciate the clarification of “censored” but do not agree it is the best way to describe what happens to those participants. Please review what you wrote in the response to the reviewer and consider if it can be better integrated in the text.

We thank the reviewer for this comment and we added an explanation in the text as suggested (lines 151-152).

Results

• I appreciate the attention provided by the reviewers toward revisions in this section but believe it would benefit from additional revisions. Regarding the differences in questionnaires and items (e.g, some were multiple choice and some were not), the authors might consider explaining this more clearly earlier on in the methods so that it can be more easily understood in the results.

RE:

We thank the reviewer for this comment and we added an explanation in the text as suggested (lines 100-101)

Discussion

• Although I appreciate the interpretation about young mothers, I want to know why this may be the case. Do you have any citations to demonstrate young mothers might have lower risk perceptions? Or any theory as to why? I know this is not the main purpose of the paper but when considering interpretations provided in the discussion, think about how to best explaining the findings in your results because that information is most helpful to the readers. You do a great job of this with the maternal psychological distress comments.

RE: We thank the reviewer for this observation. Some demographic risk factors for childhood injury were already identified in different studies and of course the maternal age is one of them. Our result about the association between maternal age and FERF is very consistent with results from other studies. In the systematic review of Mitton et al., which identified cohorts of school-aged children and adolescents, is hypothesized that younger mothers, compared with older, may be less aware of the risks that children encounter as they develop and grow (Mitton et al. 2008). We added a sentence about this in order to better explain this concept (lines 221-225)

• Lines 243-247 are very difficult to follow, please revise for clarity.

RE: We thank the reviewer for this comment and we better explain the concept (lines 247-249).

---

## [Editor Report · Decision Letter 2]

19 Sep 2022

Unintentional injuries and potential determinants of falls in young children: results from the Piccolipiù Italian birth cohort

PONE-D-21-37522R2

Dear Dr. culasso,

We’re pleased to inform you that your manuscript has been judged scientifically suitable for publication and will be formally accepted for publication once it meets all outstanding technical requirements.

Kind regards,

Angela Lupattelli, PhD

Academic Editor

PLOS ONE

---

## [Editor Report · Acceptance letter]

26 Sep 2022

PONE-D-21-37522R2 

Unintentional injuries and potential determinants of falls in young children: results from the Piccolipiù Italian birth cohort. 

Dear Dr. culasso:

I'm pleased to inform you that your manuscript has been deemed suitable for publication in PLOS ONE. Congratulations! Your manuscript is now with our production department. 

Kind regards, 

on behalf of

Dr. Angela Lupattelli 

Academic Editor

PLOS ONE